# *Lacticaseibacillus rhamnosus* dfa1 Attenuate Cecal Ligation-Induced Systemic Inflammation through the Interference in Gut Dysbiosis, Leaky Gut, and Enterocytic Cell Energy

**DOI:** 10.3390/ijms24043756

**Published:** 2023-02-13

**Authors:** Tongthong Tongthong, Warerat Kaewduangduen, Pornpimol Phuengmaung, Wiwat Chancharoenthana, Asada Leelahavanichkul

**Affiliations:** 1Department of Microbiology, Faculty of Medicine, Chulalongkorn University, Bangkok 10330, Thailand; 2Center of Excellence in Translational Research in Inflammation and Immunology (CETRII), Faculty of Medicine, Chulalongkorn University, Bangkok 10330, Thailand; 3Tropical Immunology and Translational Research Unit, Department of Clinical Tropical Medicine, Faculty of Tropical Medicine, Mahidol University, Bangkok 73170, Thailand; 4Division of Nephrology, Department of Medicine, Faculty of Medicine, Chulalongkorn University, Bangkok 10330, Thailand

**Keywords:** cecal ligation, phlegmon, dysbiosis, leaky gut, probiotics

## Abstract

Despite an uncommon condition, the clinical management of phlegmon appendicitis (retention of the intra-abdominal appendiceal abscess) is still controversial, and probiotics might be partly helpful. Then, the retained ligated cecal appendage (without gut obstruction) with or without oral *Lacticaseibacillus rhamnosus* dfa1 (started at 4 days prior to the surgery) was used as a representative model. At 5 days post-surgery, the cecal-ligated mice demonstrated weight loss, soft stool, gut barrier defect (leaky gut using FITC-dextran assay), fecal dysbiosis (increased *Proteobacteria* with reduced bacterial diversity), bacteremia, elevated serum cytokines, and spleen apoptosis without kidney and liver damage. Interestingly, the probiotics attenuated disease severity as indicated by stool consistency index, FITC-dextran assay, serum cytokines, spleen apoptosis, fecal microbiota analysis (reduced *Proteobacteria*), and mortality. Additionally, impacts of anti-inflammatory substances from culture media of the probiotics were demonstrated by attenuation of starvation injury in the Caco-2 enterocyte cell line as indicated by transepithelial electrical resistance (TEER), inflammatory markers (supernatant IL-8 with gene expression of *TLR4* and *NF-κB*), cell energy status (extracellular flux analysis), and the reactive oxygen species (malondialdehyde). In conclusion, gut dysbiosis and leaky-gut-induced systemic inflammation might be helpful clinical parameters for patients with phlegmon appendicitis. Additionally, the leaky gut might be attenuated by some beneficial molecules from probiotics.

## 1. Introduction

The intra-abdominal peritonitis using cecal ligation and puncture (CLP) in rodents is an established model of sepsis, a potentially life-threatening condition in response to a severe infection, that represents “ruptured appendicitis” or “perforated diverticulitis” in patients [1]. Indeed, acute appendicitis, resulting from obstruction of the appendiceal lumen commonly by fecaliths and lymphoid follicular hyperplasia, is a common surgical condition in humans that needs an emergency resection [2]. However, in some cases of appendicitis, the inflamed appendix is walled off by the greater omentum inducing an inflammatory tumor-liked lesion referred to as “phlegmon appendicitis” or “abscess appendicitis” which mostly does not develop sepsis and has controversial management (surgical procedures or supportive management). For surgical procedures, antibiotics administration followed by delayed abscess drainage or appendectomy versus an immediate phlegmon removal is under the debate as some cases with immediate surgery end up with an ileocecal resection due to the technical difficulty in an acute inflammatory lesion [3,4]. On the other hand, antibiotics administration followed by delayed surgery might develop sepsis or recurrent symptom with the high cost of hospital stay [4]. Meanwhile, in the cases with successful antibiotic responses, the need for delayed surgical procedures is questioned because of the low risk of recurrent inflammation [4]. Although several interventions (ultrasonogram and computed tomography) were proposed to indicate the necessity of surgical procedures after successful supportive treatment [5], an alteration in gut microbiota or gut barrier damage might be another indicator.

As such, gut dysbiosis is an imbalance of gut microbiota (gut normal flora) correlating with unhealthy outcomes, that is affected by alterations in the intra-intestinal lumen and in the systemic situation [6]. For the intra-intestinal factors, increased gut pathogens, some diets, antibiotics, and the local inflammation (infection or non-infection) [7,8,9,10] induce gut dysbiosis, in part, through the enhanced intestinal immune cells that might have a different impact on different groups of gut organisms causing selective growth in some groups of bacteria (dysbiosis) [11,12,13,14]. For systemic alterations, the selected growth of some organisms over other groups might be due to increased uremic toxin in the gut (the intestine is used as an alternative route for the toxin excretion during renal insufficiency), reduced blood perfusion in sepsis, immune responses defects, deposition of circulating immune complex [15,16,17,18,19,20], and systemic viral infection (COVID-19 and dengue) [21,22]. While gut dysbiosis from both factors damages the gut barrier (a single cell layer separating the host’s circulation and the microbial molecules in the gut contents), gut eubiosis (the balanced microbiota in the healthy regular condition) improves gut integrity [7,23], partly through the increased short-chain fatty acids (SCFAs; the growth factors for gut epithelium) that are altered from the ingested complex carbohydrates by the hosts [24]. Subsequently, the intestinal barrier defect allows the translocation of microbial molecules from the gut into the blood circulation with systemic inflammation (leaky gut or gut leakage) [25,26,27,28]. Hence, gut dysbiosis and leaky gut in phlegmon appendicitis might represent a severe extra-intestinal lumen inflammation that possibly is an indicator of the need for earlier surgical management [12].

Additionally, the use of probiotics (the microbes that are beneficial to the host) for several preventive purposes is currently mentioned and *Lacticaseibacillus rhamnosus* dfa1, isolated from the Thai healthy volunteers [29] might be beneficial for the protection of the intestine against several issues. Perhaps, phlegmon appendicitis in individuals who regularly have probiotics for the balanced gut microbiota might have some benefits once the phlegmon develops. Then, cecal ligation without puncture model was used to test the correlation between extra-intestinal lumen inflammation and intestinal conditions (leaky gut and gut dysbiosis) and the influences of probiotics. While cecal ligation and puncture (CLP) is a well-established model for sepsis [30,31,32], the information on cecal ligation model is still very scare. Indeed, ligation of the cecal appendix is an easy procedure to induce the retention of necrotic tissue inside the intra-abdominal space of mice which usually surround by the greater omentum (adipose tissue) as a mechanism for the control of infection similar to phlegmon appendicitis in patients [33]. Notably, both the human appendix (from the terminal end of the cecum) and the cecal appendage in rodents are located at the junction of the ileum and cecum; the human appendix is a narrow extension that is an evolutionary vestige, whereas the rodent cecal appendage is very large [34]. In rodents, the cecum appendix resembles a blind sac for the temporary storage of food contents allowing bacteria to break down cellulose in the contents and absorb water for the well-formed feces [35,36,37] and resection of the cecal appendage might reduce fecal SCFAs [37] and decrease energy expenditure from the reduced biomass of gut microbiota [38]. Although there is no available data on the model of cecal ligation without cecal removal in mice, this model might be a model that represents phlegmon appendicitis. We hypothesize that cecal ligation might induce gut dysbiosis and leaky gut that could possibly be attenuated by *L. rhamnosus* dfa1.

Due to (i) the lack of data on dysbiosis during inflammation at the extra-intestinal lumen without sepsis or any underlying condition, (ii) the possible clinical translation in the patients with phlegmon appendicitis, and (iii) the interest in the cecal ligation model as a model representing a human condition, a model of cecal ligation with or without the probiotics together with several in vitro experiments was performed.

## 2. Results

### 2.1. The Clinical Manifestation and Fecal Dysbiosis of Mice with Cecal Ligation

To induce intraperitoneal inflammation and to test the impact of probiotics on the model, cecal ligation was performed with or without probiotic administration. As such, all mice with cecal ligation demonstrated weight loss that was more prominent in cecal-ligated mice without probiotics (Figure 1A). Although there was a low mortality rate in cecal-ligated mice, the operated mice with probiotics showed a higher 7-day survival rate (Figure 1B). There were looser and softer feces (stool consistency index) in the cecal-ligated mice when compared with the sham control (Figure 1C,D), possibly due to the water reabsorption through the mouse cecum. However, the cecal-ligated mice with probiotics demonstrated a better-formed stool than the non-probiotic mice at 2 days (but not at 5 days) post-surgery (Figure 1C,D). Although no puncture was performed in the ligated cecum, bacteremia was detectable in some mice mostly at 5 days post-surgery (Figure 1E). In parallel, the cecal-ligated mice demonstrated gut barrier defect, as evaluated by FITC-dextran assay (Figure 1F), as early as 2 days post-surgery, despite mostly negative hemoculture (Figure 1E), compared with the control mice without the significant difference to the 5 days post-operation. Although the probiotic administration reduced leaky gut (FITC-dextran) at 2 days post-surgery, the values at 5 days were not different between groups (Figure 1F). On the other hand, elevation of serum TNF-α was detected as early as 1 day after cecal ligation, whereas serum IL-6 and IL-10 increased on the 2nd and 5th day post-operation (Figure 1G–I). Probiotics attenuated serum cytokine levels at 2 days post-surgery (all cytokines) and at 5 days post-operation (serum TNF-α and IL-6) (Figure 1G–I). While the increased serum cytokines were not high enough to induce liver damage (liver enzyme) (Figure 1J), the systemic inflammation showed some impacts on the spleen as indicated by spleen apoptosis (Figure 1K). Indeed, the abundance of spleen apoptosis in the probiotic-administered mice was lower than in the cecal-ligated mice without probiotics (Figure 1K and Figure 2), which might be correlated with the attenuation of leaky-gut-induced systemic inflammation by probiotics (Figure 1F–G).

Because (i) both leaky gut and stool consistency might be partly associated with gut dysbiosis [39], (ii) mouse cecum is an important reservoir of gut organisms [37], and (iii) the possible vulnerability of the balance of gut organisms [40], cecal ligation might interfere with fecal dysbiosis. As such, in comparison with control mice, cecal-ligated mice demonstrated a higher abundance of bacteria in phylum *Proteobacteria* (the group of several pathogenic Gram-negative bacteria), especially *Escherichia-Shigella* (Figure 3A–C) with a reduction in the diversity of bacterial species as evaluated by Chao1 richness estimation (the total number of species in the population) and Shannon evenness score (the proportions of species of the microbiota) (Figure 3D).

In comparison with the sham control, the crude characteristics of fecal microbiota from the cecal-ligated mice were demonstrated by the abundance of bacterial phylum (Figure 3C). Indeed, there was lower *Firmicutes*, the group of potentially beneficial bacteria in the healthy host [41], with the higher *Proteobacteria* (also referred to as “*Pseudomonadota*”), the group of several pathogenic Gram-negative bacteria [42], and similar *Bacteroidota*, the group of mostly Gram-negative anaerobes with possible pathogenicity in some situation [43]. Bacteria in Phylum *Desulfobacterota*, the thermophilic sulfate-reducing bacteria [44], and *Patescibacteria*, the uncultivated bacteria from metagenomic analysis [45], also presented in the highest abundance in the sham control mice (Figure 3A–C), but with a very limited amount of data on impacts of these organisms in the host. Nevertheless, the use of probiotics reduced *Proteobacteria* and increased *Bacteroides* without the alteration in *Firmicutes* and bacterial diversity (Figure 3A–D). Additionally, the differences in fecal microbiome among these experimental groups were also indicated by the clear separation in the non-metric multidimensional scaling (NMDS; a statistical analysis allowing the complex multivariate data sets to be visualized in a reduced number of dimensions) (Figure 3E).

### 2.2. The Protective Effects of the Condition Media from Lacticaseibacillus rhamnosus dfa1 in Enterocytes

Because (i) the excretion of several beneficial substances from *Lacticaseibacillus rhamnosus* dfa1, especially the exopolysaccharide is well-known [46,47] and (ii) cell starvation might be the most common and most important underlying mechanism of enterocyte damages caused by several different insults (such as hypoxia, overwhelming inflammatory responses, and some toxins) [48,49,50], the in vitro starvation model using an enterocyte cell line (Caco-2 cell) and the Lacticaseibacillus condition media (LCM) might be an interesting experiment to test the general protective effect of probiotics. As such, with 48 h starvation protocol with or without LCM (see method), there was no alteration in the cell viability as evaluated by the MTT assay when compared with the control group (Figure 4A). Surprisingly, the 48 h starvation enhanced the enterocyte integrity, as measured by transepithelial electrical impedance (TEER), when compared with the control (Figure 4B). However, the starvation caused enterocytic pro-inflammatory responses as indicated by increased supernatant cytokine (IL-8) with the up-regulation of Toll-like receptor 4 (*TLR-4*) and nuclear factor kappa B (*NF-κB*) (Figure 4C–E). For the cell energy status, the starvation decreased mitochondrial function (oxygen consumption rate; OCR) without an alteration in glycolysis activity (extracellular acidification rate; ECAR) (Figure 4F–I), despite the starvation-induced enterocyte inflammatory responses (Figure 4C–E), supporting a previous publication [51]. Additionally, the cell starvation also induced oxidative stress as indicated by an increase in a reactive oxygen species (ROS) using malondialdehyde (MDA) measurement (Figure 4J), possibly due to mitochondrial injury-induced oxidative stress [52].

In starvation with LCM, the cell viability was not altered but enterocyte integrity (TEER) was enhanced with a reduction in inflammatory responses (supernatant IL-8 and gene expression of *TLR4* and *NF-κB*) (Figure 4A–E), implying some beneficial impacts of probiotics. Notably, in Caco-2 cells with starvation and LCM, the expression of *NF-κB* was reduced to the level of the control group (Figure 4E); however, the anti-inflammatory property of LCM was not high enough to normalize supernatant IL-8 and *TLR4* expression (Figure 4C,D). In the cell energy status, LCM reduced both mitochondrial function (OCR) and glycolysis activity (ECAR) along with decreased oxidative stress (MDA) (Figure 4F–J). Perhaps, the reduced cell energy status might be correlated with the anti-inflammatory cell activity that reduced mitochondrial injury and ROS production. Although exploration of the mechanistic details of probiotics is out of the scope of the current interest, our data suggest that *L. rhamnosus* dfa1 attenuates disease severity of the mice with cecal ligation partly through the improved gut dysbiosis and excretion of some anti-inflammatory substances that strengthen gut barrier integrity partly via cell energy interference (Figure 5).

## 3. Discussion

For the characteristics of the model, necrosis of the mouse cecal appendage resulted in a significant weight loss, possibly due to reduced intake and poor water reabsorption (more fecal fluid loss). Indeed, the rodent cecal appendage is similar to appendix in human that possibly be used to solidify the rodent feces [35,36,37]. Some cecal-ligated mice died as early 2 days post-operation possibly due to bacteremia; however, bacteremia was detectable only in a few mice per group (especially at 5 days post-operation) resulting in the non-statistically significant value of bacteremia among the experimental groups. Despite a low rate of bacteremia in cecal-ligated mice, the intestinal barrier defect in these mice as determined by the translocation of FITC-dextran (molecular weight 4 kDa), approximately 4 × 10^−3^ µm in diameter (smaller than the size of the whole viable bacteria at 1–2 µm in diameter) [53,54], from the gut into the blood circulation (leaky gut) was demonstrated. Likewise, systemic inflammation (low levels of the elevated serum cytokines) in cecal-ligated mice was also demonstrated which might be due to the immune responses against the intra-abdominal necrotic tissue [55,56] together with the leaky gut-induced systemic inflammation [57]. Different from the cecal ligation and puncture (CLP) sepsis model which gut perforation and fecal peritonitis are initiated by a puncture of the ligated cecum, mice with cecal ligation without puncture, here, show only a subtle systemic inflammation with relatively low mortality compared with the high mortality rate in CLP sepsis model [58,59,60]. The limited mortality in cecal-ligated mice supported the role of greater omentum in the control of intra-abdominal sources of infection (abscess or necrotic tissue) as reported in phlegmon appendicitis [5]. Although the systemic inflammation after cecal ligation was not severe enough to induce kidney or liver injury, different from the inflammation of the gut–kidney–liver axis in sepsis [7], the significant inflammatory responses in cecal-ligated mice were demonstrated by an increase in apoptosis of spleen. Indeed, the overwhelming inflammatory activation can cause apoptosis in several cell types is well-known [27,61].

While serum cytokines in cecal-ligated mice might be a result of both leaky gut and other factors, fecal dysbiosis in these mice supported a possible dysbiosis-induced leaky gut after cecal ligation surgery. As such, an increased *Proteobacteria* (mostly pathogenic Gram-negative bacteria) and reduced bacterial diversity in the feces of cecal-ligated mice support gut dysbiosis in this model. With a presence of intra-abdominal necrotic tissue in this model, the inflammation in the peritoneal cavity affects the balance of gut microbiota, despite the inflammatory lesions being walled off. Although these data are similar to the dysbiosis in several peritonitis models, such as spontaneous bacterial peritonitis, fecal peritonitis, and peritonitis in peritoneal dialysis [62,63,64], the dysbiosis in these situations might not be only from peritonitis alone but also from the underlying overt systemic responses, including cirrhosis, sepsis, and uremia, in these situations that might affect the gut dysbiosis. Here, despite the subtle systemic responses in the cecal ligation model, the alteration in fecal dysbiosis was prominent, supporting a vulnerability of the balance in microbiota in the gut to the microenvironment [65]. Indeed, the alteration in psychological conditions, environmental changes, and physical stresses can also affect gut microbiota [66,67] and the detection of dysbiosis might be one of the sensitive biomarkers to recognize some important clinical impacts in patients. Indeed, the interplay between gut dysbiosis and inflammation is well-known [68]; however, data on the correlation between a subtle systemic inflammation and gut dysbiosis without the additional impacts from the underlying disease are still very limited. Hence, the cecal ligation (without puncture) model might be one of the interesting models for the exploration of dysbiosis with an inactive intra-abdominal lesion (abscess or necrotic tissue).

For the probiotic impacts, the stool bulk-forming property [34] and the enhanced intestinal integrity by probiotics are well-known through several mechanisms, including direct impacts on enterocytes (SCFAs; the energy sources of intestinal cells) [69], immune modulation (such as exopolysaccharides) [70,71], and reduced gut pathogens (nutrient competition) [72]. Here, probiotics attenuated dysbiosis after cecal ligation as indicated by reduced *Proteobacteria*, but not good enough to enhance bacterial diversity. Probiotics also improved gut barrier defect (FITC-dextran assay) and decreased serum cytokines in cecal-ligated mice, supporting that elevated serum cytokines in these mice might be, at least in part, correlated with leaky gut. In the in vitro experiments, starvation protocol on enterocytes was used as a model for inducing cell injury because cell starvation might be a common factor of cell injury in several stimulations. Although there was a good adaptation of enterocytes toward several insults (such as hypoxia and lipopolysaccharide) [73,74,75], the overwhelming inflammation induces cell injury, partly through the use-up of cell energy and mitochondrial injury [76]. Likewise, hypoxia, cellular acidosis, and poor blood perfusion (from too much vasodilation or cardiac dysfunction) are also similar to cell starvation as these factors limit energy production [77]. As expected, cell starvation reduced the mitochondrial function of enterocytes (extracellular flux analysis) which might be correlated with an increase in oxidative stress (MDA). Notably, mitochondria are important sources of oxidants in the cells, partly because of components in the electron transport chain [78], that can transform lipids in cell membrane into MDA [79]. However, the starvation had a low impact on glycolysis activity, perhaps due to the lower cell energy production (the synthesis of adenosine triphosphate; ATP) by glycolysis compared with mitochondrial oxidative phosphorylation [80].

With the condition media of *Lacticaseibacillus rhamnosus* dfa1 (LCM), there were improved enterocyte integrity (TEER), reduced inflammatory responses, decreased cell energy status (both mitochondrial function and glycolysis activity), and decreased oxidative stress in starvation-induced enterocyte injury. While the starvation injury reduced cell energy with elevated cytokine production and enhanced oxidative stress implying the starvation-induced mitochondrial injury [81], the prominently lower energy status from probiotics induced lower inflammatory responses with less oxidative stress suggesting enterocyte protection [82]. Potentially, the necessity of energy production from mitochondria and glycolysis is reduced by the short-chain fatty acids (SCFAs) in LCM that can be metabolized mostly by enterocytes (and hepatocytes) [83]. It is well-known that the presence of SCFAs inhibits the glycolysis pathway, reduces equivalents of the mitochondrial respiratory chain, decreases the efficacy of oxidative ATP synthesis, and enhances lipogenesis and gluconeogenesis [84,85]. More mechanistic studies will be of interest.

Although our current study is only a proof of concept on gut dysbiosis induction by the inflammation in the intraperitoneal cavity, the cecal ligation model might be a good representative model of phlegmon appendicitis [4]. Because leaky gut, fecal dysbiosis, and systemic inflammation are the main characteristics that correlate with mortality rate, we propose using these factors (or one of these factors) as the indicator for the removal of intra-abdominal phlegmon lesions. For example, patients with phlegmon appendicitis who develop endotoxemia (an indicator of leaky gut) might need an urgent phlegmon removal, whereas the patients without endotoxemia might benefit more from conservative antibiotic treatment. More studies on this model and in patients are needed for testing this hypothesis. For the use of probiotics, our current results implied that the probiotics might prevent the severity of dysbiosis caused by the retained intra-abdominal abscess; however, the use of probiotics in the patients might be inappropriate as the avoidance of oral intake will be routinely performed. Nevertheless, regular intake of probiotics for other health benefits might be also protective against intra-peritoneal abscess-induced dysbiosis and systemic inflammation. Additionally, isolation of the active anti-inflammatory substances from probiotics might be interesting.

Finally, there were several limitations in our study. First, the animal model study with a limited number of mice in each group is only a proof of concept study that need further experiments for a solid conclusion. Second, the gender and age of the mice might affect the model as our current study only uses young male 8-week-old mice that represent young humans (approximately 20 years old) [86]. Studies on female mice of different ages might have different results. Third, more details in mechanistic parts are needed for understanding the role of ligated cecal appendage on gut dysbiosis and the working mechanisms of probiotics. Despite these limitations, the cecal ligation (without puncture) model might be another model representing some conditions of the patients that might be useful for translation research. 

## 4. Materials and Methods

### 4.1. Animal Model and Mouse Sample Analyses

The animal study protocol (CU-ACUP No. 011/2564) was endorsed by the Institutional Animal Care and Use Committee of the Faculty of Medicine, Chulalongkorn University following the U.S. National Institutes of Health (NIH) animal care and use protocol. As such, C57BL/6J mice were purchased from Nomura Siam (Pathumwan, Bangkok, Thailand) and 8-week-old male mice were used in the experiments. All mice were housed in an animal facility under a light/dark cycle of 12:12 h at 22 ± 2 °C with 50 ± 10% relative humidity with free access to water and food (SmartHeart Rodent; Perfect companion pet care, Bangkok, Thailand). Then, *Lacticaseibacillus rhamnosus* dfa1 (Chulalongkorn University, Bangkok, Thailand) that isolated from the Thai healthy volunteers from a previous project [29] at 1 × 10^8^ CFU in 0.3 mL phosphate-buffered solution (PBS), or PBS alone, were daily orally administered for 4 days prior to cecal ligation or sham operation. The cecal ligation protocol was modified from previous publications [87,88,89,90,91,92]. Briefly, the whole length of the cecum was ligated by silk 2-0 before closing the abdominal wall layer by layer, whereas sham surgery was only the identification of the cecum before suturing the abdominal wall with nylon 6–0. After that, 1 mL of prewarmed normal saline solution (NSS) with tramadol at 25 mg/kg/dose was subcutaneously administered after surgery, at 6 and 18 h post-CLP. The oral gavage by *L. rhamnosus* dfa1 or PBS was omitted on the day of surgery and continued after that. All mice were observed and monitored daily with body weight measurement. For blood and organ sample collection, the mice were euthanized by cardiac puncture under isoflurane anesthesia. The serum was stored at −80 °C until use, the spleen was placed in 10% neutral formalin for histological analysis, and feces were collected for microbiome analysis. The stool consistency index was graded into four scores as follows; 0: normal, 1: soft, 2: loose, and 3: diarrhea followed previous publications [93,94]. Blood bacterial abundance (bacteremia) was evaluated using the direct spread of mouse blood onto blood agar plates (Oxoid, Hampshire, UK) in serial dilutions and incubating at 37 °C for 24 h before colony enumeration.

For leaky gut evaluation, 0.5 mL of 25 mg/mL fluorescein isothiocyanate (FITC)-dextran 4 kDa (Sigma-Aldrich, St. Louis, MO, USA) in sterile water was orally administered at 3 h prior to sacrifice and FITC-dextran in blood was measured by fluorospectrometer (Varioskan, Thermo Fisher Scientific, Foster City, CA, USA) relative to FITC-dextran standards as previously described [10,15,16]. Notably, the results of FITC-dextran from different time points were determined from the different mice due to the retention of FITC-dextran in the intestinal lumens for a few days after administration. Serum cytokines (TNF-α, IL-6, and IL-10) and alanine transaminase (the liver enzyme) were measured by the enzyme-linked immunosorbent assay (ELISA) (Invitrogen, Waltham, MA, USA) and EnzyChrom Alanine Transaminase assay (EALT-100) (BioAssay, Hayward, CA, USA). Serum creatine (kidney function) was evaluated by QuantiChrom (DICT-500, BioAssay). The spleen samples on 4 mm paraffin sections were stained by hematoxylin and eosin (H&E) color and by an immunohistochemistry stain using an anti-activated caspase-3 antibody (Cell Signaling Technology, Beverly, MA, USA). Apoptotic cells in the spleen were evaluated in 10 randomly selected ×200 magnified fields per slide and expressed as positive cells per high-power field following previous publications [27,95,96,97].

### 4.2. Fecal Microbiome Analysis

Mouse feces were collected by placing mice in metabolic cages (Hatteras Instruments, Cary, NC, USA) for a few hours. Then, feces (0.25 g) from each mouse in different cages were collected for microbiome analysis to avoid the influence of allocoprophagy (a habit of mice that ingest feces from other mice) on fecal microbiota analysis following previous publications [93,98,99]. Briefly, the metagenomic DNA was extracted from the prepared samples using a DNAeasy Kit (Qiagen, Redwood City, CA, USA). The quality and concentration of the extracted DNA were measured by nanodrop spectrophotometry. Universal prokaryotic primers 515F (5′-GTGCCAGCMGCCGCGGTAA-3′) and 806R (5′-GGACTACHVGGGTWTCTAAT-3′) with appended 50 Illumina adapter and 30 Golay barcode sequences were used for 16S rRNA gene V4 library construction in Miseq300 platform (Illumina, San Diego, CA, USA). The raw sequences and operational taxonomic unit (OTU) were classified following Mothur’s Standard Operating Procedures (SOP) [100].

### 4.3. The In Vitro Experiments

Caco-2 (HTB-37), human colorectal adenocarcinoma cells from the American Type Culture Collection (ATCC, Manassas, VA, USA), were cultured with Dulbecco’s modified Eagle medium (DMEM) supplemented with 20% fetal bovine serum (FBS), 1% penicillin/streptomycin, 4-(2-hydroxyethyl)-1-piperazineethanesulfonic acid (HEPES) with sodium pyruvate (Thermo Fisher Scientific) in a humidified 5% CO_2_ incubator at 37 °C. Then, the cells were sub-cultured before using for the starvation protocol by transferring the cells into the serum-free culture medium (DMEM supplemented with 1% penicillin/streptomycin, 1% HEPES, and 1% sodium pyruvate) for 48 h following a previous protocol [73]. The regular culture media were used as a control group. Meanwhile, the condition media (modified DMEM as mentioned above) after 48 h culture with *L. rhamnosus* dfa1, referred to as “Lacticaseibacillus condition media (LCM)”, was prepared by the incubation of *L. rhamnosus* dfa1 at an OD_600_ of 0.1 in anaerobic condition for 48 h before supernatant collection by centrifugation and filtration with a 0.22 μm membrane filter (Minisart; Sartorius Stedim Biotech GmbH, Göttingen, Germany). After that, this cell-free supernatant (0.5 mL) was concentrated by speed vacuum drying at 40 °C for 3 h (Savant Instruments, Farmingdale, NY, USA) and stored at −20 °C until use [101]. Then, the LCM pellets, resuspended in DMEM (5% LCM) or DMEM alone (Media), in an equal volume were incubated in the Caco-2 cells with starvation before the sample collection for supernatant cytokines by ELISA (Invitrogen) or gene expression with quantitative real-time polymerase chain reaction (PCR) following previous publications [102,103,104]. 

Briefly, the total RNA was prepared by Trizol, quantified by a Nanodrop ND-1000 (Thermo Fisher Scientific, Rockford, IL, USA), converted to cDNA with High-Capacity cDNA Reverse Transcription (Thermo Scientific), and examined gene expression with the SYBR Green PCR Master Mix (Applied biosystem, Foster City, CA, USA). The results were demonstrated in relative quantification of the comparative threshold method (2^−ΔΔCt^) and normalized by a housekeeping gene (*β-actin*). The following primers (for human cells) were used to amplify cDNA fragments: *TLR4,* forward 5′-CACAGACTTGCGGGTTCTAC-3′, reverse 5′-AGGACCGACACACCAATGATG-3′; *NF-κB*, forward 5′-CAGAGCTGCGCTTGCAGAG-3′, reverse 5′-GTCAGCAGCCGGTTACCAAG-3′; *β-actin* forward 5′-CCTGGCACCCAGCACAAT-3′, reverse5′-GCCGATCCACACGGAGTACT-3′. In parallel, the cell viability was determined by tetrazolium dye 3-(4,5-dimethylthiazol-2-yl)-2,5-diphenyltetrazolium (MTT) solution (Thermo Fisher Scientific) [13] with the incubation by 0.5 mg/mL of MTT solution at 37 °C in the dark for 2 h and diluted by dimethyl sulfoxide (DMSO; Thermo Fisher Scientific) before measurement with a Varioskan Flash microplate reader at absorbance OD570 nm. Moreover, the impact of starvation on intestinal integrity was determined by transepithelial electrical resistance (TEER) as previously described [19]. Briefly, Caco-2 cells at 5 × 10^4^ cells per well in modified DMEM were seeded onto the upper compartment of the 24-well Boyden chamber trans-well for 15 days to establish the confluent cell monolayer. Then, the starvation protocol was performed and TEER in ohm (Ω) × cm2 was measured with an epithelial volt-ohm meter (EVOM2TM, World precision instruments, Sarasota, FL, USA) by placing electrodes in the supernatant at basolateral and in apical chambers. Notably, TEER values in media culture without Caco-2 cells were used as a blank and were subtracted from the values for the measurements. For measurement of malondialdehyde (MDA; an indicator of reactive oxygen species of lipid peroxidation) [105], the activated Caco-2 cells were homogenized by the Ultra-Turrax homogenizer (IKA, Staufen, Germany) and centrifuged at 12,000× *g* for 15 min at 4 °C to separate the supernatant. Then, malondialdehyde (MDA) in the supernatant was measured by an MDA assay kit (colorimetric) (Abcam, Cambridge, UK) according to the manufacturer’s protocol and representing the intracellular reactive oxygen species (ROS).

### 4.4. Extracellular Flux Analysis

Seahorse XFp Analyzers (Agilent, Santa Clara, CA, USA) was used to determine the energy status of the cells (extracellular flux analysis), with oxygen consumption rate (OCR) and extracellular acidification rate (ECAR) representing mitochondrial function (respiration) and glycolysis activity, respectively, follow previous publications [17,88,91,106,107]. Briefly, Caco-2 cells (1 × 10^4^ cells/well) were grown in Seahorse culture plates with DMEM for 48 h before transferring to the serum-free culture medium (starvation protocol) or control DMEM for another 48 h with or without 5%LCM. Then, the cell media were replaced by Seahorse media (DMEM complemented with glucose, pyruvate, and L-glutamine) (Agilent, 103575–100) at 37 °C for 1 h and subsequently activated by different metabolic interference compounds, including 1.5 μM oligomycin, 1 μM carbonyl cyanide-4-(trifluoromethoxy)-phenylhydrazone (FCCP), and 0.5 μM rotenone/antimycin A for the oxygen consumption rate (OCR) evaluation. The maximal respiration was calculated by the Seahorse Wave 2.6 software based on the following equation, maximal respiration = OCR between FCCP and rotenone/antimycin A—OCR after rotenone/antimycin A. Additionally, glycolysis stress tests were calculated from the mitochondrial stress test using the wave program of Seahorse XF Analyzers (Agilent) and demonstrated by the area under the curve of the ECAR graph as calculated by the trapezoidal rule [91]. 

### 4.5. Statistical Analysis

All data in mean ± standard error (SE) were analyzed by Graph Pad Prism version 7.0 software (La Jolla, CA, USA) and Statistical Package for Social Sciences software (SPSS 22.0, SPSS Inc., Chicago, IL, USA). The differences between multiple groups were examined by one-way analysis of variance (ANOVA) with Tukey’s analysis, whereas the survival analysis was determined by the log-rank test. A *p*-value < 0.05 was considered statistically significant.

## 5. Conclusions

The vulnerability of the balance of fecal organisms was demonstrated by gut dysbiosis after cecal ligation. Gut dysbiosis and leaky-gut-induced systemic inflammation could be demonstrated in some mice which might be possibly used as biomarkers for patients with an intra-abdominal abscess as an indication of the severe disease. In addition, the reduced severity of cecal-ligated mice with probiotic prevention also supports the possible benefit of balanced eubiosis on the reduction of systemic inflammation and disease severity. More studies are warranted.

## Figures and Tables

**Figure 1 ijms-24-03756-f001:**
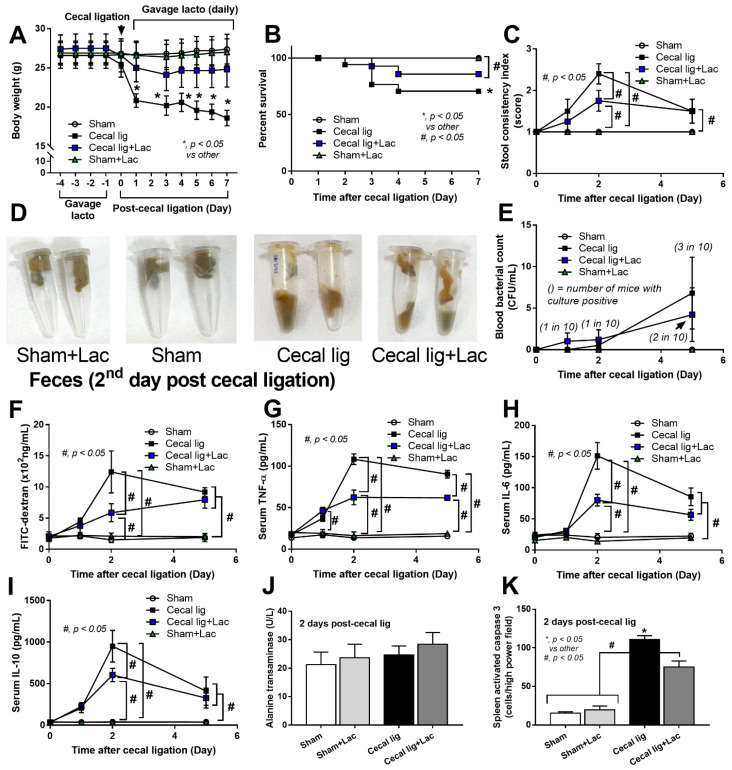
The characteristics of mice with cecal ligation (without puncture) (Cecal lig) or sham surgery (Sham) with or without Lacticaseibacillus administration (Cecal lig + Lac and Sham + Lac) as indicated by body weight (**A**), survival analysis (**B**), stool consistency index (**C**), representative pictures of the mouse feces (**D**), blood bacterial count (**E**), gut barrier determination (FITC-dextran assay) (**F**), serum cytokines (TNF-α, IL-6, and IL-10) (**G**–**I**), liver enzyme (alanine transaminase) (**J**), and spleen apoptosis (**K**) are demonstrated (n = 25/group in A and B, n = 8–10/group or time point for others).

**Figure 2 ijms-24-03756-f002:**
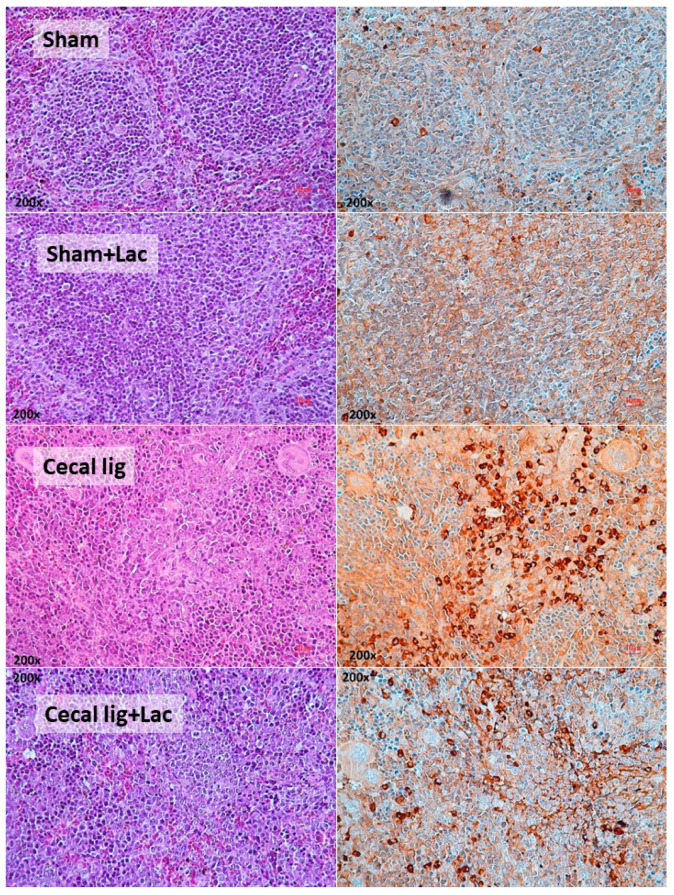
The representative pictures of spleen histology from mice with cecal ligation (without puncture) (Cecal lig) or sham surgery (Sham) with or without Lacticaseibacillus administration (Cecal lig + Lac and Sham + Lac) at 48 h post-surgery as stained by Hematoxylin and Eosin (H&E) stain (left column) and activated caspase 3 (apoptosis) (right column) are demonstrated (the score of apoptotic cells is presented in Figure 1K).

**Figure 3 ijms-24-03756-f003:**
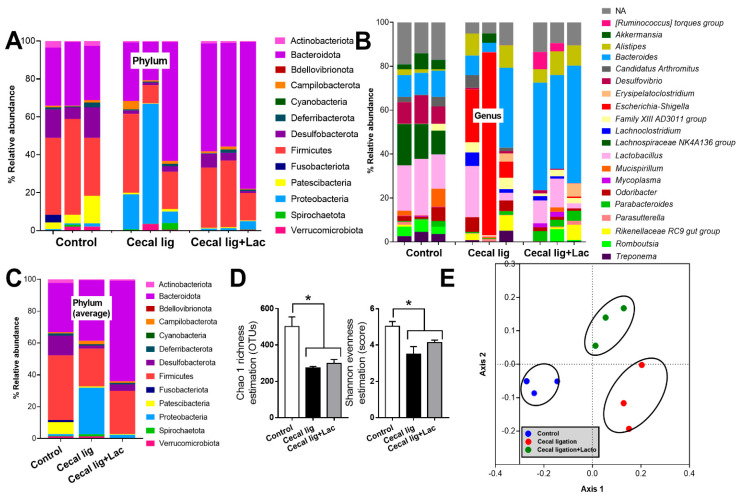
The characteristics of fecal microbiome analysis from mice with cecal ligation (without puncture) (Cecal lig) or sham surgery (Sham) with or without Lacticaseibacillus administration (Cecal lig + Lac and Sham + Lac) as indicated by the abundance of bacteria in phylum, genus, and the average value on phylum (**A**–**C**), the alpha diversity (Chao 1 richness estimation and Shannon evenness evaluation) (**D**), and the non-metric multidimensional scaling (NMDS) (**E**) are demonstrated. * *p* < 0.05.

**Figure 4 ijms-24-03756-f004:**
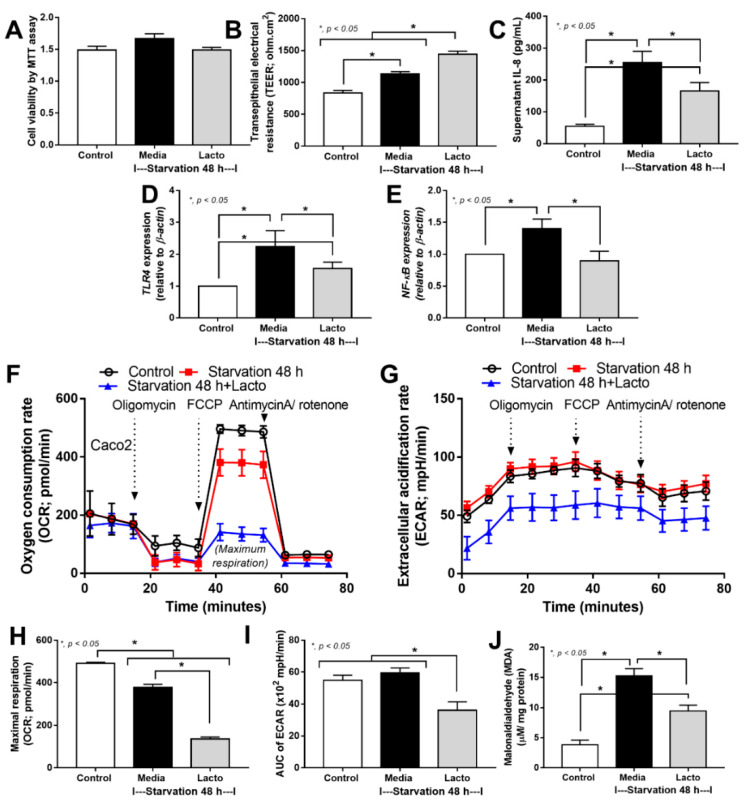
The characteristics of enterocytes (Caco-2 cell line) in control media versus the 48 h starvation with or without Lacticaseibacillus condition media (LCM) (Lacto) as indicated by cell viability test using 3-(4,5-dimethylthiazol-2-yl)-2,5-diphenyl-2H-tetrazolium bromide (MTT assay) (**A**), enterocyte integrity (transepithelial electrical resistance; TEER) (**B**), supernatant IL-8 (**C**), gene expression of inflammatory molecules, including Toll-like receptor 4 (*TLR4*), nuclear factor kappa B (*NF-κB*) (**D**,**E**), cell energy status as evaluated by the oxygen consumption rate (OCR; mitochondrial function), extracellular acidification rate (ECAR; glycolysis activity), maximal respiration (calculated from OCR), and the area under the curve (AUC) of ECAR (**F**–**I**), with malondialdialdehyde (MDA; a reactive oxygen species) (**J**) are demonstrated (independent triplicated experiments were performed).

**Figure 5 ijms-24-03756-f005:**
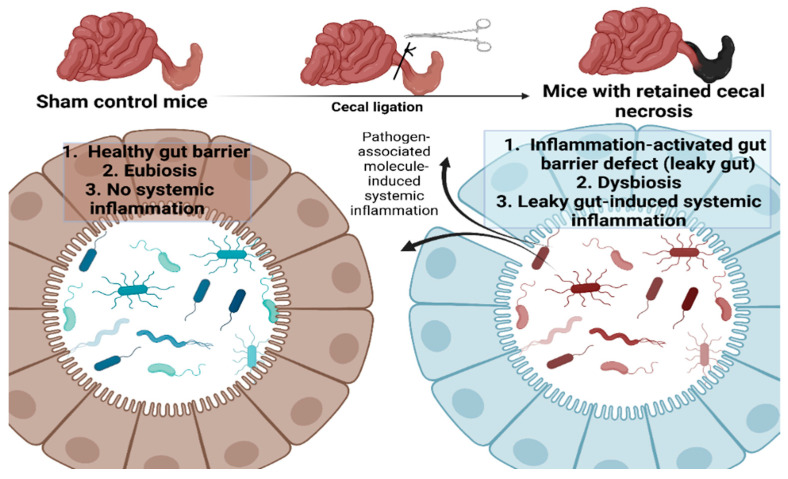
The proposed working hypothesis of the gut barrier defect (leaky gut) from the retained necrotic lesion in the abdomen possibly through the active intraperitoneal inflammation that induces fecal dysbiosis (prominence in *Proteobacteria*; the phylum of several Gram-negative pathogenic bacteria) leading to the translocation of microbial molecules from the gut into the blood circulation (leaky gut) that facilitating systemic inflammation is demonstrated. Picture created by BioRender.com.

## Data Availability

Not applicable.

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
