# Peer review of "Lacticaseibacillus rhamnosus dfa1 Attenuate Cecal Ligation-Induced Systemic Inflammation through the Interference in Gut Dysbiosis, Leaky Gut, and Enterocytic Cell Energy"

_ijms, 2023, doi:10.3390/ijms24043756_

Round 1
Reviewer 1 Report
It is an interesting and important research question, and the authors performed a detailed set of experiments.
The paper overall tells the story; however, at many points, careful interpretation of findings should be made to avoid misleading, and authors should avoid interpreting findings in the results section.
No methodological flaws. The manuscript needs to be rewritten for clarity and interpretation of findings. Would recommend avoiding terms like "few", and "some" throughout the manuscript and rewriting the manuscript using appropriate numbers, percentages, group mean, median, variance, and appropriate p-values, along with the tests used to generate those p-values. It will be helpful to proofread the entire manuscript for grammar mistakes.
In the discussion, the authors should mention the study's limitations in a paragraph.

Author Response
Authors have addressed most if not all comments and suggestions raised by the reviewers and amended the manuscript accordingly. Then, it has certainly been improved.
However, there was a clear message from Reviewer 1 that was not taken into consideration: the Abstract section should be completely rewritten, from scratch. Everything is obscure there; thus, sentences should be short, clear, and ordered in a logical way one after another, following background, experimental conditions, main results, and conclusions.
ANS: We apologize for our mistake and rewrite all part of the abstract in the new version of manuscript.
Also important is the fact that only a single strain has been used in these experiments, which means that the results are bound to this specific strain. Therefore, the complete name of the strain (genus, species, strain) should clearly state in the title.
ANS: We agree with the editor and correct it accordingly.
Further, the name of the species Lactobacillus rhamnosus was replaced in the Abstract was replaced by Lacticaseibacillus rhamnosus, which agrees with the updated nomenclature, but not at other positions of the manuscript.
ANS: We apologize for our mistake and replace the name in all part of the new version manuscript.

Reviewer 2 Report
Dear Authors,
your manuscript entitled "Lactobacilli attenuate cecal ligation-induced systemic inflammation through the interference in gut dysbiosis, leaky gut, and enterocytic cell energy" is interesting because of its scope of finding a treatment alternative to surgery to treat phlegmon appendicitis, i.e. use of probiotics, by using cecum ligation in mice as a model. However, the manuscript must be improved in clarity and presentation as suggested in the uploaded annotaded pdf.

Author Response

(The authors gave the same response as above.)
